# TOWARDS MULTIMODAL ACTIVE LEARNING: EFFICIENT LEARNING WITH LIMITED PAIRED DATA

## ABSTRACT

Active learning (AL) is a principled strategy to reduce annotation cost in data-hungry deep learning. However, existing AL algorithms focus almost exclusively on single-modality data, overlooking the substantial annotation burden in multimodal learning. We introduce the first framework for *multimodal active learning with unaligned data*, where the learner must actively acquire cross-modal alignments rather than labels on pre-aligned pairs. This setting captures the practical bottleneck in modern multimodal pipelines such as CLIP and SigLIP, where unimodal features are easy to obtain but high-quality alignment is costly. We develop a new algorithm that combines uncertainty and diversity principles in a modality-aware design, achieves linear-time acquisition, and applies seamlessly to both pool-based and streaming-based settings. Extensive experiments on benchmark datasets demonstrate that our approach consistently reduces multimodal annotation cost while preserving performance; for instance, on the ColorSwap dataset it cuts annotation requirements by up to $40\%$ without loss in accuracy.

## 1 INTRODUCTION

Deep learning has achieved remarkable success across a wide range of applications, but its effectiveness often hinges on access to large amounts of annotated training data. Active learning (AL) has long been viewed as a promising approach to reduce annotation cost by selectively querying the most *informative* instances for labeling (Settles, 2009). Theoretically, AL can *exponentially* reduce the amount of labeled data required (Zhu & Nowak, 2022a), and empirically, it has delivered consistent gains in data efficiency for deep models (Sener & Savarese, 2017; Ash et al., 2019; Citovsky et al., 2021; Saran et al., 2023; Zhang et al., 2024a), as well as more recently for large language models (Margatina et al., 2023; Bhatt et al., 2024; Yuan et al., 2025).

Despite this progress, most existing AL methods focus on the *unimodal* setting with *unidirectional annotation*: given unlabeled features, the learner queries class labels from a small, fixed set. The most closely related multimodal extension is the *pre-aligned* multimodal AL setting introduced in Shen et al. (2023), which *assumes* that vision-language pairs are *already aligned* and simply queries labels on these pairs. Since the alignment is free, the problem effectively reduces to unimodal AL on composite inputs rather than addressing the harder challenge of discovering cross-modal correspondences. Other applications of AL to multimodal tasks, such as video captioning (Zhang et al., 2024b), follow a similar pattern: one modality (e.g., video) is treated as unlabeled input and the other (e.g., text) directly as annotation, making the process unimodal in nature.

In this paper, we introduce the first setting of *multimodal active learning with unaligned data*, where the learner begins with independent vision and language features and must actively acquire cross-modal alignments. Unlike unimodal AL or pre-aligned multimodal AL, our setting requires deciding both *which modality to query from* and *how to align instances across modalities*. This formulation raises two qualitatively new challenges: (i) *bidirectional alignment*, since annotation can begin from either vision-to-language or language-to-vision, and different choices lead to distinct annotation sets and learning trajectories; and (ii) a *large cross-modal candidate space*, since evaluating the utility of an instance requires reasoning over potential matches across the entire other modality, which may contain millions of unique candidates (Gadre et al., 2024). Naively scoring all image–text pairs scales quadratically, making classical AL strategies computationally infeasible.

Our setting is directly motivated by modern multimodal pipelines such as CLIP (Radford et al., 2021) and SigLIP (Zhai et al., 2023), where raw modality-specific features can be obtained cheaply at scale, but high-quality alignment is expensive, domain-specific, and often the true bottleneck (Gadre et al., 2024; Bai et al., 2024). This challenge is especially acute in specialized domains such as medical imaging (Chen & Hong, 2024) and autonomous driving (Ge et al., 2023), where multimodal annotation is both costly and critical.

**Our contributions.** Our main contributions are as follows:

(i) We introduce the problem of *multimodal active learning with unaligned data*, clarifying how it differs fundamentally from unimodal AL and existing multimodal AL with pre-aligned data.

(ii) We develop a new algorithm that integrates uncertainty and diversity principles in a modality-aware design, achieves *linear-time* complexity in the number of unaligned instances, and applies seamlessly to both *pool-based* and *streaming-based* scenarios.

(iii) We conduct extensive experiments on benchmark datasets, demonstrating consistent annotation savings (up to $40\%$ on ColorSwap) while maintaining competitive performance.

**Paper organization.** Section 2 introduces our problem formulation and highlights the unique challenges of multimodal AL with unaligned data. Section 3 presents our algorithm, its complexity analysis, and extensions. Section 4 reports the main experimental results, and Section 5 provides additional analysis. We conclude in Section 6. Related work and additional details are deferred to the Appendix due to space constraints.

## 2 PROBLEM SETTING

We study multimodal learning with a dataset $\mathcal{D} = (\mathcal{D}^v, \mathcal{D}^l)$, where $\mathcal{D}^v = \{x_i^v\}_{i=1}^n$ denotes the collection of raw vision features and $\mathcal{D}^l = \{x_i^l\}_{i=1}^n$ denotes the collection of raw textual/language features.[1] Unlike standard multimodal setups, *the vision and language features are initially unaligned*. The learner may query a subset of instances to obtain their aligned pairs at an annotation cost. Specifically, for any data point $x_i^k \in \{x_i^v, x_i^l\}$, the learner can spend one unit of *annotation cost* to reveal its aligned pair $x_i := (x_i^v, x_i^l)$. We use $\mathcal{S} = \{(x_i^v, x_i^l)\}_{i=1}^m$ to denote the set of annotated pairs obtained with a total of $m$ units of cost.

The goal, under a fixed annotation budget, is to *actively and strategically* select an informative subset $\mathcal{S}$ to maximize the quality of a multimodal model $\phi := (\phi^v, \phi^l)$, where $\phi^v, \phi^l : \mathbb{R}^{d'} \to \mathbb{R}^d$ are encoders that map raw features into a shared representation space. We adopt CLIP-style contrastive training for multimodal models (Radford et al., 2021), and, following standard practice (Zhai et al., 2022; 2023), evaluate model quality on downstream tasks.

We refer to this setup as *multimodal active learning with unaligned data*, which not only extends classical unimodal active learning (Sener & Savarese, 2017; Ash et al., 2019; Citovsky et al., 2021; Saran et al., 2023) but also departs fundamentally from prior multimodal active learning frameworks that assume *pre-aligned* data (Shen et al., 2023). The active learning algorithm proceeds over $T \in \mathbb{Z}_+$ iterations. At iteration $t$, the learner selects and annotates a batch of $B$ data points $\{(x_{t_i}^v, x_{t_i}^l)\}_{i=1}^B$, and updates the annotation set as $\mathcal{S}_t \leftarrow \mathcal{S}_{t-1} \cup \{(x_{t_i}^v, x_{t_i}^l)\}_{i=1}^B$. The multimodal model $\phi_t = (\phi_t^v, \phi_t^l)$ is trained on $\mathcal{S}_t$ and then used to guide data selection in the next iteration. This iterative process enables the learner, under a fixed budget, to build a high-quality multimodal model from strategically chosen alignments.

Depending on how the learner accesses the unaligned pool, we study two regimes:

- **Pool-based multimodal active learning.** The learner has full access to $\mathcal{D} = (\mathcal{D}^v, \mathcal{D}^l)$ throughout the process and can query any instances at any $t \in [T]$ (Ash et al., 2019).

- **Streaming-based multimodal active learning.** The pool arrives as disjoint subsets $\mathcal{D} = \{\mathcal{D}_t\}_{t=1}^T$ with $\cup_{t=1}^T \mathcal{D}_t = \mathcal{D}$ and $\mathcal{D}_{t_i} \cap \mathcal{D}_{t_j} = \emptyset$. At iteration $t$, the learner only observes $\mathcal{D}_t = (\mathcal{D}_t^v, \mathcal{D}_t^l)$

---

[1]For simplicity, we focus on the vision-language case. Our setting and algorithms naturally extend to general multimodal learning with more than two modalities; see Section 3.2.

and must select and annotate data *within* this batch. Unqueried data from $\mathcal{D}_t$ cannot be revisited in future iterations (Saran et al., 2023).

**Additional notation.** For any $N \in \mathbb{Z}_+$, we denote $[N] := \{1, \cdots, N\}$. For multimodal sets $\mathcal{D} = \{\mathcal{D}^v, \mathcal{D}^l\}$ and $\mathcal{S} = \{\mathcal{S}^v, \mathcal{S}^l\}$, we write $\mathcal{D} \setminus \mathcal{S} := \{\mathcal{D}^v \setminus \mathcal{S}^v, \mathcal{D}^l \setminus \mathcal{S}^l\}$. For any modality $k \in \{v, l\}$, we denote $\phi_t^k(\mathcal{S}_t^k) := \{\phi_t^k(x^k) : x^k \in \mathcal{S}_t^k\}$. When clear, we use the shorthand $\phi_t(\mathcal{S}_t^k) = \phi_t^k(\mathcal{S}_t^k)$ or $\phi(\mathcal{S}_t^k) = \phi_t^k(\mathcal{S}_t^k)$.

## 2.1 UNIQUE CHALLENGES WITH UNALIGNED MULTIMODAL DATA

Compared to unimodal active learning (Sener & Savarese, 2017; Ash et al., 2019) and multimodal active learning with *pre-aligned* data (Shen et al., 2023), our setting—multimodal active learning with *unaligned* data—poses qualitatively new challenges.

In unimodal active learning, each instance $x$ has a single feature vector and annotation assigns a class label from a small predefined set. In multimodal active learning with *pre-aligned* data, the learner selects from pre-aligned pairs $\{(x_i^v, x_i^l)\}_{i=1}^n$ for label queries. Because modalities are already aligned, this effectively reduces to unimodal active learning on composite inputs.

By contrast, in our unaligned setting with vision features $\{x_i^v\}_{i=1}^n$ and language features $\{x_i^l\}_{i=1}^n$, the learner must simultaneously decide *which modality to query from* and *how to align instances across modalities*. This creates two distinctive challenges:

- **Bidirectional alignment.** With unaligned data, annotation may begin from either vision-to-language or language-to-vision. Crucially, the learner does not know in advance which instance from the other modality will be paired. Different alignment directions can lead to entirely different annotation sets, adding an extra decision layer absent in unimodal or pre-aligned multimodal AL.

- **Large cross-modal candidate space.** Even after choosing a modality to query, the utility of a candidate must be evaluated against the entire other modality. For instance, querying an image requires scoring potential matches across all texts, which effectively serves as an enormous candidate label space. Unlike conventional class labels, these candidates are instance-specific and extremely numerous (e.g., 12.8M unique texts in DataComp (Gadre et al., 2024)). Naively evaluating all pairs scales quadratically with dataset size, quickly becoming infeasible.

Thus, multimodal active learning with *unaligned* data requires acquisition algorithms that handle both bidirectional alignment and the large cross-modal search space efficiently. We present our approach to these challenges in Section 3.

## 3 METHODS

We present our approaches to address the unique challenges mentioned in Section 2.1. We first introduce our multimodal active learning algorithm for the pool-based setting in Section 3.1, which is further extended to the streaming-based setting and to settings beyond vision-language models in Section 3.2.

### 3.1 MUTLIMODAL ACTIVE LEARNING

We present our multimodal active learning algorithm in Algorithm 1, which proceeds iteratively for $T$ iterations. At each iteration $t \in [T]$, Algorithm 1 selects a batch of $B$ data points for annotation, based on the multimodal model $\phi_{t-1} = (\phi_{t-1}^v, \phi_{t-1}^l)$ trained with respect to previously annotated data points $\mathcal{S}_{t-1}$. The annotation set is then updated to $\mathcal{S}_t$, and the model $\phi_t = (\phi_t^v, \phi_t^l)$ is retrained on the updated annotation dataset to guide data selection in the next iteration.

The core of Algorithm 1 lies in how to select the batch of $B$ data points for annotation. This is achieved via three integrated steps: (1) modality selection, (2) coreset construction, and (3) uncertainty-based selection. At a high level, Algorithm 1 first selects a modality that is *under-represented* by the already annotated data $\mathcal{S}_{t-1}$ (Step 1). It then constructs a coreset of $B_C$ data on the selected modality to ensure *coverage* (Step 2), and finally selects $B \leq B_C$ highly *uncertain*

data points from this coreset using a cross-modal uncertainty score (Step 3), which quantifies how confidently a feature in one modality matches candidates from the other.

---

**Algorithm 1** Multimodal Active Learning

---

**Input:** Unaligned multimodal dataset $\mathcal{D} = \{\mathcal{D}^v, \mathcal{D}^l\}$, number of iterations $T$, per-round selection size $B$, coreset hyperparameter $B_C \geq B$.
1: Initialize multimodal model $\phi_0 = \{\phi_0^v, \phi_0^l\}$ with random or pretrained weights.
2: Initialize the annotation set $\mathcal{S}_0 = \emptyset$.
3: **for** $t = 1, \cdots, T$ **do**
4:    Consider unaligned data pool $\mathcal{D}_t := \mathcal{D} \setminus \mathcal{S}_{t-1}$.
5:    **Step 1: Modality selection.** Select modality $k_t := \arg\max_{k \in \{v,l\}} d_t^k$ that is less-covered by previous annotations $\mathcal{S}_{t-1}$, where

$$d_t^k := \max_{z_i \in \bar{\phi}_{t-1}(\mathcal{D}_t^{k_t})} \min_{z_j \in \bar{\phi}_{t-1}(\mathcal{S}_{t-1}^{k_t})} \mathsf{dist}(z_i, z_j). \tag{1}$$

6:    **Step 2: Coreset construction.** On modality $k_t$, construct a coreset $\mathcal{C}_t^{k_t} \subseteq \mathcal{D}_t^{k_t}$ of size $B_C$ such that, together with $\mathcal{S}_{t-1}^{k_t}$, it maximally covers $\mathcal{D}_t^{k_t}$:

$$\mathcal{C}_t^{k_t} := \arg\min_{\mathcal{C}:|\mathcal{C}|=B_C} \max_{z_i \in \phi_{t-1}(\mathcal{D}_t^{k_t} \setminus \mathcal{C})} \min_{z_j \in \phi_{t-1}(\mathcal{S}_{t-1}^{k_t} \cup \mathcal{C})} \mathsf{dist}(z_i, z_j). \tag{2}$$

   // Eq. (2) can be approximated with an efficient greedy algorithm (Algorithm 2).
7:    **Step 3: Uncertainty-based selection.** Within coreset $\mathcal{C}_t^{k_t}$, select the top-$B$ *most uncertain* data points using multimodal model $\phi_{t-1}$.
8:    Let $m := \{v, l\} \setminus \{k_t\}$ denote the unselected modality in Step 1.
9:    For each data $x_i^{k_t} \in \mathcal{C}_t^{k_t}$, compute its margin score $u(x_i^{k_t}) := w_{(1)}^i - w_{(2)}^i$, which serves as an uncertainty measure. Here $w^i \in \mathbb{R}^{|\mathcal{D}_t^m|}$ denotes the vector of similarity scores between $x_i^{k_t}$ and all unaligned features in the other modality $\mathcal{D}_t^m$, and $w_{(j)}^i$ denotes the $j$-th largest entry of $w^i$. // Compute margin score as an uncertainty measure.
10:    Select the subset $\{x_i^{k_t}\}_{i=1}^B \subseteq \mathcal{C}_t^{k_t}$ with the top-$B$ uncertainty scores (i.e., lowest margins), annotate them, and update $\mathcal{S}_t \leftarrow \mathcal{S}_{t-1} \cup \{(x_i^v, x_i^l)\}_{i=1}^B$.
11:    **Model update.** Train multimodal model $\phi_t = (\phi_t^v, \phi_t^l)$ on the updated annotation set $\mathcal{S}_t$.
**Output:** Actively trained multimodal model $\phi_T = (\phi_T^v, \phi_T^l)$.

---

Our Algorithm 1 integrates *both diversity and uncertainty principles* into multimodal active learning with unaligned data. By restricting cross-modal uncertainty evaluation to the coreset constructed in Step 2, our algorithm enables *efficient data selection* with per-round runtime that scales *linearly* in $|\mathcal{D}|$. In contrast, a naive uncertainty-based selection over the entire dataset would require *quadratic* time in $|\mathcal{D}|$, which is computationally prohibitive.

We next explain the details of each of the three steps in Algorithm 1.

**Step 1: Modality selection.** This step selects the modality that is *underrepresented* with respect to the current annotation $\mathcal{S}_{t-1}$ (line 5 in Algorithm 1). To assess coverage, for each modality $k \in \{v, l\}$, we compute the maximum distance of unaligned features $\mathcal{D}_t^k := \mathcal{D}^k \setminus \mathcal{S}_{t-1}^k$ to their nearest neighbors in $\mathcal{S}_{t-1}^k$:

$$d_t^k := \max_{z_i \in \bar{\phi}_{t-1}(\mathcal{D}_t^{k_t})} \min_{z_j \in \bar{\phi}_{t-1}(\mathcal{S}_{t-1}^{k_t})} \mathsf{dist}(z_i, z_j),$$

where dist is a distance metric, and $\bar{\phi}(x)$ denotes normalized embeddings to ensure comparability across modalities. The modality with the largest distance value (i.e., least covered), $k_t := \arg\max_{k \in \{v,l\}} d_t^k$, is selected for coreset construction in the next step. This step has a runtime upper bound of $O(|\mathcal{D}_t| \cdot |\mathcal{S}_{t-1}|)$.

**Step 2: Coreset construction.** Given the selected modality $k_t$, in Step 2 (line 6 of Algorithm 1), we construct a coreset $\mathcal{C}_t^{k_t} \subseteq \mathcal{D}_t^{k_t}$ of size $B_C \geq B$ that, when combined with already annotated data

points $\mathcal{S}_{t-1}^{k_t}$, maximally covers the unaligned data $\mathcal{D}_t^{k_t}$. This is formalized in Eq. (2) of Algorithm 1. Solving Eq. (2) exactly is NP-Hard (Cook et al., 1994). Following prior work in diversity-based active learning in the unimodal setting (Sener & Savarese, 2017), we employ a greedy approximation algorithm (Algorithm 2) that guarantees a $2 \times \mathsf{OPT}$ solution with runtime $O(B_C \cdot |\mathcal{D}_t| \cdot |\mathcal{S}_{t-1}|)$.

**Step 3: Uncertainty-based selection.** With the constructed coreset $\mathcal{C}_t^{k_t}$, in Step 3 (lines 7-10 of Algorithm 1), we compute the margin score as the a measure of uncertainty and select the top-$B$ data points with the highest uncertainty (i.e., lowest margin scores) for multimodal annotation. For each $x_i^{k_t} \in \mathcal{C}_t^{k_t}$, we compute a vector of similarity scores $w^i \in \mathbb{R}^{|\mathcal{D}_t^m|}$ between $x_i^{k_t}$ and all unaligned features in the other modality $m := \{v, l\} \setminus \{k_t\}$. The calculation of similarity score is usually model-dependent; but a simple example is the inner product between representations (or its variants), e.g., $w_j^i := \langle \phi_{t-1}^{k_t}(x_i^{k_t}), \phi_{t-1}^m(x_j^m) \rangle$ for $x_j^m \in \mathcal{D}_t^m$. The uncertainty score is computed as the margin between the top two similarity scores: $u(x_i^{k_t}) := w_{(1)}^i - w_{(2)}^i$. We select the $B$ data points in $\mathcal{C}_t^{k_t}$ with the *lowest* margin scores. This step has a runtime of $O(B_C \cdot |\mathcal{D}_t|)$.

---

**Algorithm 2** Greedy Approximation for Coreset Construction

---

**Input:** Selected modality $k_t$, unaligned multimodal dataset $\mathcal{D}_t^{k_t}$, selection size $B_C$, multimodal model $\phi_{t-1}$, current annotation set $\mathcal{S}_{t-1}$.
1: Initialize coreset set $\mathcal{C}_t^{k_t} = \emptyset$.
2: **while** $|\mathcal{C}_t^{k_t}| < B_C$ **do**
3:     Select a data point $z_u$ such that:
$$z_u := \underset{z_i \in \phi_{t-1}(\mathcal{D}_t^{k_t})}{\arg\max} \min_{z_j \in \phi_{t-1}(\mathcal{S}_{t-1}^{k_t})} \mathsf{dist}(z_i, z_j).$$
4:     Update $\mathcal{C}_t^{k_t} \leftarrow \mathcal{C}_t^{k_t} \cup \{z_u\}$.
**Output:** Coreset $\mathcal{C}_t^{k_t}$.

---

**Computational complexity.** Summing the runtime of Steps 1–3, the per-round data acquisition complexity of Algorithm 1 is upper bounded by $O(B_C \cdot |\mathcal{D}_t| \cdot |\mathcal{S}_{t-1}|)$, where $|\mathcal{S}_{t-1}| = O(tB)$ and $|\mathcal{D}_t| \leq |\mathcal{D}|$.[2] Across $T$ rounds, the total complexity is upper bounded by $O(T^2 \cdot B \cdot B_C \cdot |\mathcal{D}|)$, as formalized in Proposition 1. The dominant factor is $O(|\mathcal{D}|)$, which is typically large in real-world multimodal learning tasks. In contrast, a naive uncertainty-based approach would compute margin scores for *all pairs* across modalities in the unaligned data, resulting in a per-round complexity of $O(|\mathcal{D}|^2)$, which is computationally expensive. Our algorithm avoids this quadratic bottleneck by computing cross-modal uncertainty *only over the coreset (Step 3)*. We present the computational complexity analysis in Proposition 1 and defer the formal proof to Appendix A.2.

**Proposition 1.** *The per-round data acquisition complexity of Algorithm 1 is upper bounded by* $O(B_C \cdot |\mathcal{D}_t| \cdot |\mathcal{S}_{t-1}|)$, *resulting in an overall complexity of* $O(T^2 \cdot B \cdot B_C \cdot |\mathcal{D}|)$.

## 3.2 Extensions of Algorithm 1

**Streaming-based multimodal active learning.** While Algorithm 1 is originally designed for the pool-based setting, it can be easily adapted to streaming-based multimodal active learning. As introduced in Section 2, in the streaming-based setting, the learner only has access to the current batch of stream data $\mathcal{D}_t$. To adapt Algorithm 1 to this setting, we simply replace line 4 with the current batch of stream data $\mathcal{D}_t$, and leave all other parts of the algorithm unchanged. As a result, aside from potential variation in the size of $\mathcal{D}_t$, the per-round computational complexity remains same as in the pool-based case.

**Active learning beyond vision-language models.** Although our focus in this paper is on multimodal learning with vision-language data, Algorithm 1 can be naturally extended to general mul-

---

[2]We focus on the data acquisition complexity and its dependency on the size of the data pool $O(|\mathcal{D}|)$, which is usually large for multimodal learning; note that the model training complexity is the same for all active learning algorithms and it's proportional to the training data size $O(|\mathcal{S}_{t-1}|)$.

Table 1: Datasets used in our experiments and their corresponding evaluation metrics.

| Datasets | Evaluation metrics | #Data samples |
|---|---|---|
| ColorSwap | Scores: text, image, group (Burapacheep et al., 2024) | 1400 |
| MS-COCO | R@1: I $\to$ T, T $\to$ I (Zhai et al., 2023) | 118K |
| DataComp | Average score over 38 tasks (Gadre et al., 2024) | 12.8M |

timodal settings with $m \geq 3$ unaligned modalities $\mathcal{D} = \{\mathcal{D}^{(1)}, \mathcal{D}^{(2)}, \cdots, \mathcal{D}^{(m)}\}$. To support this generalization, we modify Step 1 and Step 3 of the algorithm while keeping Step 2 unchanged, since it operates solely within the selected modality. We discuss these changes below:

- For Step 1, we generalize the modality selection in Eq. (1) to select the least-covered modality *among all $m$ modalities*, i.e., $k_t := \arg\max_{k \in \{1,2,...,m\}} d_t^k$.

- For Step 3, given any data point $x^{k_t} \in \mathcal{D}_t^{k_t}$ in the selected modality, let $u^j(x^{k_t})$ denote its cross-modal uncertainty score with respect to each unselected modality $j \in [m] \setminus \{k_t\}$. We then define the *overall uncertainty score* as $u(x^{k_t}) := \sum_{j \in [m] \setminus \{k_t\}} u^j(x^{k_t})$, and use this score for uncertainty-based selection.

Assuming $m = O(1)$, the computational complexity of this generalized algorithm remains order-wise the same as the analysis provided in Proposition 1.

## 4 EXPERIMENTS

We conduct extensive experiments to evaluate the effectiveness of our proposed algorithm. The experimental setup is described in Section 4.1, followed by the main results and analyses in Section 4.2. We defer additional implementation details and experimental results to Appendix A.3.

### 4.1 EXPERIMENTAL SETUPS

**Datasets.** We conduct experiments on three multimodal datasets: ColorSwap (Burapacheep et al., 2024), MS-COCO (Lin et al., 2014), and DataComp (Gadre et al., 2024). ColorSwap is designed to evaluate object-color matching with color-swapped image-caption pairs. MS-COCO is a large-scale image-caption dataset designed for object detection. DataComp consists of large-scale image-text pairs collected from Common Crawl. See Table 1 for detailed descriptions of these datasets. We use ColorSwap for pool-based AL, and MS-COCO and DataComp for streaming-based AL. For ColorSwap and MS-COCO, we initialize models from pretrained weights and study multimodal active learning in the *finetuning* regime. For DataComp, we initialize models from random weights to evaluate multimodal active learning in the *pretraining* regime.

**Baselines and models.** We compare our algorithm against three baselines: RAND, CORESET, and UNCERTAINTY. RAND serves as a passive learning baseline, randomly selecting data pairs for annotation. Since multimodal active learning with unaligned data is a new problem, no existing baselines directly apply. To better assess the performance of our Algorithm 1, we construct two additional baselines by adapting widely used unimodal methods to the multimodal setting: a diversity-based method (CORESET) and an uncertainty-based method (UNCERTAINTY).[3]

We implement our algorithm and all baselines using the CLIP model (Radford et al., 2021) and its variants SigLIP (Zhai et al., 2023) and LiT (Zhai et al., 2022), evaluating across models of different sizes. Detailed hyperparameter settings are reported in Appendix A.3.2.

**Evaluation metrics.** We adopt standard evaluation metrics for each dataset. For ColorSwap, we report the text score, image score, and group score, as proposed by Burapacheep et al. (2024). For MS-COCO, we report recall@1 for both image-to-text and text-to-image retrieval, as commonly used in the literature (Zhai et al., 2023). For DataComp, we follow Gadre et al. (2024) and report

---

[3]Full algorithmic details of CORESET and UNCERTAINTY are provided in Appendix A.3.1.

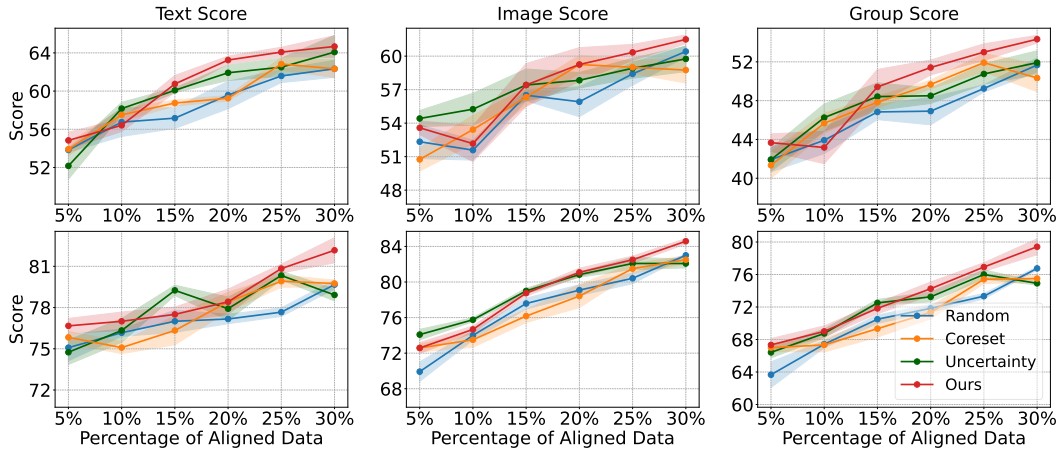

Figure 1: Results of pool-based multimodal active learning on the ColorSwap dataset with CLIP-B32 (*top*) and SigLIP-B16 (*bottom*). We report text score (*left*), image score (*middle*), and group score (*right*) as learning progresses.

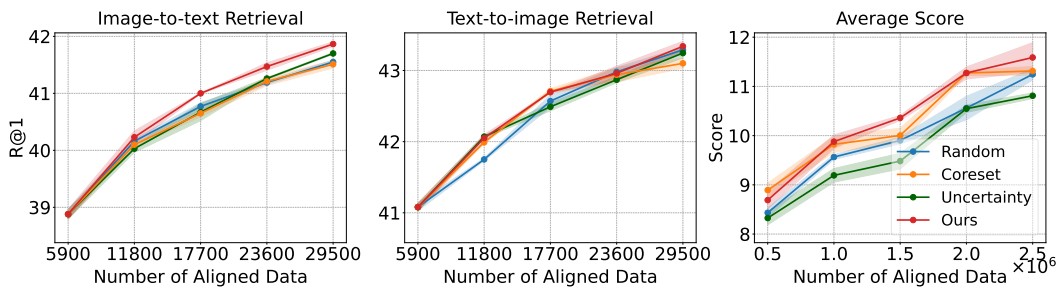

Figure 2: Streaming-based multimodal active learning with the MS-COCO (*left and middle*) and DataComp (*right*) datasets using CLIP-B32. We report R@1 (image-to-text) (*left*), R@1 (text-to-image) (*middle*), and the average score across 38 downstream tasks (*right*). We report algorithm performance as learning progresses.

the average score across 38 downstream tasks. All results are averaged over 4 random runs, with shaded regions in plots indicating $2/3$ of a standard deviation.

## 4.2 MAIN RESULTS

**Pool-based multimodal active learning.** We compare Algorithm 1 against three baselines on the ColorSwap dataset. As shown in Fig. 1, our algorithm generally outperforms the baselines across all three metrics. Notably, with CLIP-B32, Algorithm 1 achieves a group score of $49.42$ using only $15\%$ of the data, which is comparable to the group score reached by RAND at $25\%$, corresponding to a $40\%$ reduction in annotation cost. Relative to UNCERTAINTY, our algorithm achieves a group score of $51.42$ with $20\%$ of the data, while UNCERTAINTY requires $30\%$ to reach a similar score—representing a $33\%$ reduction in cost. In addition to superior data efficiency, Algorithm 1 is computationally more efficient than UNCERTAINTY, which incurs higher complexity (Section 3.1).

**Streaming-based multimodal active learning.** We next evaluate Algorithm 1 in the streaming-based setting on MS-COCO and DataComp datasets. On MS-COCO (left and middle panels of Fig. 2), our algorithm consistently outperforms all baselines across both retrieval metrics. For example, on R@1 (image-to-text), Algorithm 1 achieves a score of $41.47$ using 23,600 samples ($20\%$ of the data), matching the performance of CORESET and RAND with $25\%$ of the data. This corresponds to a $20\%$ reduction in annotation cost.

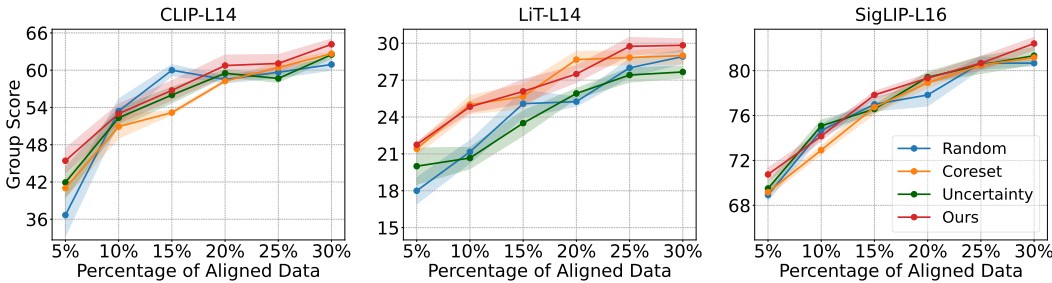

Figure 3: Group scores on the ColorSwap dataset in the pool-based setting, using CLIP-L14 (*left*), LiT-L14 (*middle*), and SigLIP-L16 (*right*).

Table 2: Ablation study of Algorithm 1 with different modality selection strategies in the pool-based setting using CLIP-B32 and SigLIP-B16. We report group scores with $30\%$ of aligned data.

| Model | Random | Text-only | Image-only | OURS |
|---|---|---|---|---|
| CLIP-B32 | $52.4_{\pm 3.7}$ | $52.8_{\pm 1.4}$ | $50.7_{\pm 2.3}$ | $\mathbf{54.3_{\pm 1.2}}$ |
| SigLIP-B16 | $75.50_{\pm 2.1}$ | $77.67_{\pm 2.9}$ | $76.89_{\pm 1.9}$ | $\mathbf{79.42_{\pm 2.9}}$ |

To examine performance at larger scales, we further conduct experiments on the DataComp dataset (right panel of Fig. 2), reporting the average score across 38 downstream tasks. Our algorithm outperforms all baselines, with particularly large margins over RAND and UNCERTAINTY. Relative to CORESET, Algorithm 1 achieves clear improvements as the number of aligned pairs grows, reaching a score of 11.59 with 2.5M pairs. For context, training on the full 12.8M aligned pairs (the performance skyline) yields a score of 13.20 (Gadre et al., 2024). Thus, our method attains $87.80\%$ of the skyline with just 2.5M pairs, whereas CORESET reaches only $85.68\%$ of the skyline (with a score of 11.31).

**Robustness across CLIP variants and model sizes.** To evaluate the robustness of Algorithm 1 across architectures and model sizes, we conduct additional experiments with multiple CLIP variants, including SigLIP (Zhai et al., 2023) and LiT (Zhai et al., 2022), as well as models of different sizes. Given the high runtime cost of the streaming-based setting, most experiments are performed in the pool-based scenario. Figure 3 reports group scores for large-scale variants, where Algorithm 1 consistently outperforms the baselines. Results for text and image scores are deferred to Appendix A.3.3, and show similar trends.

## 5 ANALYSES AND ABLATIONS

**Modality selection.** We first assess the effectiveness of the modality selection strategy in Algorithm 1 (Step 1). Following the same experimental setup as in Fig. 1, we compare our approach against three alternatives: randomly choosing a modality, always selecting the text modality, and always selecting the image modality. As reported in Table 2, our proposed strategy achieves the best performance across both CLIP and SigLIP models. For example, it attains a group score of 54.3 with the CLIP model, representing up to a $7\%$ improvement over the image-only strategy, with all other components held fixed.

**Robustness to coreset hyperparameter $B_C$.** To examine robustness to the coreset hyperparameter $B_C$ (Step 2), we conduct experiments across model families (CLIP and SigLIP) and scales (CLIP-B32 and SigLIP-L16). As shown in Fig. 4, performance remains stable across different values of $B_C$ and annotation costs, demonstrating that our method is robust to this parameter. In practice, we select the value of $B_C$ from the set $\{1.5B, 2B, 2.5B\}$, depending on the dataset and model. We provide detailed hyperparameter selections in Appendix A.3.2.

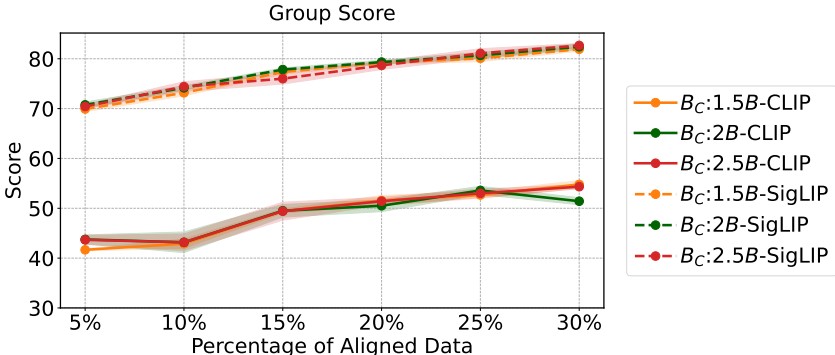

Figure 4: Parameter study of Algorithm 1 with different values of $B_C$ in the pool-based setting using CLIP-B32 and SigLIP-L16. We report group scores as learning progresses.

Table 3: Case study of Algorithm 1, recording margin scores at Step 3 under the pool-based setting with CLIP-B32. We report average margin scores (scaled by $\times 10^{-2}$) for both correctly and incorrectly matched groups (with respect to the ground-truth alignments) across different percentages of aligned data pairs.

| Percentage of Aligned Data | 5% | 10% | 15% | 20% | 25% | 30% |
|---|---|---|---|---|---|---|
| Correctly matched group | $3.8_{\pm 0.2}$ | $4.0_{\pm 0.1}$ | $4.2_{\pm 0.4}$ | $4.7_{\pm 0.2}$ | $4.8_{\pm 0.2}$ | $5.2_{\pm 0.2}$ |
| Incorrectly matched group | $0.9_{\pm 0.1}$ | $1.0_{\pm 0.2}$ | $1.3_{\pm 0.1}$ | $1.4_{\pm 0.2}$ | $1.4_{\pm 0.1}$ | $1.5_{\pm 0.5}$ |

**Effectiveness of margin score in data selection.** To evaluate the effectiveness of the margin score in Algorithm 1 (Step 3), we analyze its behavior across different percentages of aligned data pairs. At each iteration, Step 3 computes pseudo-alignments for all unaligned data points by selecting their most likely match based on similarity. We then partition the data into correctly matched and incorrectly matched groups (with respect to ground-truth alignments) and report their average margin scores in Table 3. As expected, incorrectly matched samples consistently exhibit lower margin scores than correctly matched ones, reflecting higher uncertainty and thus greater value for active selection. Since these pseudo-alignments are computed on unaligned data not yet used in training, the results confirm that the margin score provides a robust and meaningful uncertainty signal even in noisy, unaligned settings.

**Synergy of uncertainty and diversity in data selection.** As shown in Section 4.2, Algorithm 1 consistently outperforms all baselines. We attribute this advantage to its ability to prioritize data points that are both *uncertain* (Step 3) and *diverse* (Step 2). To test this hypothesis, we visualize the ColorSwap dataset using t-SNE on image-modality embeddings, comparing data selected by Algorithm 1 with those chosen by UNCERTAINTY. As shown in Fig. 8 (deferred to Appendix A.3.4), points selected by Algorithm 1 (red stars) not only concentrate near high-uncertainty regions (blue circles; selected by UNCERTAINTY) but also spread more broadly across the embedding space, reflecting greater diversity. This joint emphasis on uncertainty and diversity provides a key advantage of Algorithm 1.

# 6 CONCLUSION

We presented the first study of active learning in multimodal settings *with unaligned data*, addressing the key challenges of bidirectional alignment and annotation across large cross-modal candidate spaces. By integrating uncertainty- and diversity-based selection in a modality-aware design, we developed an efficient algorithm applicable to both pool-based and streaming-based scenarios. Experiments on benchmark datasets show that our approach reduces annotation requirements by up to 40% while maintaining model performance, highlighting the promise of active learning for scalable and cost-effective multimodal learning. Although our focus has been on multimodal representation learning, an important future direction is to extend these ideas to multimodal generative models.

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

# A APPENDIX

## A.1 RELATED WORK

**Active learning.** Active learning (AL) aims to train accurate models with fewer annotations by selectively querying the most informative instances (Settles, 2009). It has become increasingly important in modern applications where unlabeled data are abundant but annotation is costly. Theoretically, a long line of work has established the provable benefits of AL over standard passive learning (Castro & Nowak, 2007; Balcan et al., 2007; Dasgupta et al., 2009; Hanneke, 2014; Krishnamurthy et al., 2019; Puchkin & Zhivotovskiy, 2021; Zhu & Nowak, 2022b;a). Empirically, AL has also demonstrated consistent gains, especially when integrated with deep neural networks (Sener & Savarese, 2017; Ash et al., 2019; Citovsky et al., 2021; Ash et al., 2021; Wang et al., 2022; Saran et al., 2023; Zhang et al., 2024a), and more recently with large pretrained models (Margatina et al., 2023; Bhatt et al., 2024; Yuan et al., 2025).

Despite these successes, most work has focused on the *unimodal* setting with *unidirectional annotation*: given unlabeled features, the learner selects a subset for labeling. The closest multimodal extension is the *pre-aligned* setting (Shen et al., 2023), which assumes that vision-language pairs are already aligned and simply queries labels on these pairs. Since alignment is free, this effectively reduces to unimodal AL on composite examples rather than tackling the harder challenge of discovering cross-modal correspondences. Other applications of AL to multimodal tasks, such as video captioning (Zhang et al., 2024b), follow a different path: one modality (e.g., video) is treated as input while the other (e.g., text) serves as annotation, keeping the process strictly unidirectional and thus still structurally unimodal.

In contrast, we introduce the first multimodal AL setting that supports *bidirectional alignment* with unaligned data: the learner is provided with independent vision and language features and must actively acquire meaningful cross-modal correspondences, either from images to text or from text to images. This setting is directly motivated by modern multimodal pipelines such as CLIP (Radford et al., 2021) and SigLIP (Zhai et al., 2023), where unimodal features are easy to obtain at scale, but high-quality alignment is expensive, domain-specific, and often the true bottleneck.

**Multimodal learning.** Multimodal learning seeks to integrate information from diverse modalities such as text, images, and audio to improve learning performance (Baltrušaitis et al., 2018; Liang et al., 2024). Early approaches relied on supervised labels, where multimodal features were annotated with classification labels, making the process closely resemble standard unimodal learning. More recently, multimodal learning has shifted toward supervision from paired multimodal data, where one modality provides a supervision signal for another (Zong et al., 2024). A prominent example is CLIP (Radford et al., 2021) and its variants (Zhai et al., 2022; 2023), which leverage paired image–text data to train contrastive objectives that align representations across modalities.

While large-scale, noisily aligned multimodal data can be scraped from the web, it is increasingly recognized that training high-performing multimodal models requires high-quality, well-aligned datasets (Gadre et al., 2024; Bai et al., 2024). This need is even more pronounced in specialized domains such as medical imaging (Chen & Hong, 2024) and autonomous driving (Ge et al., 2023), where careful multimodal annotation is both critical and costly. These challenges highlight the importance of developing efficient methods that can learn effectively from fewer paired examples. Although recent work has explored active learning for multimodal tasks with label annotations (Shen et al., 2023), to the best of our knowledge, our work is the first to design a multimodal active learning algorithm specifically tailored for *pairing annotation*.

**Data selection.** Data selection is closely related to active learning, aiming to construct a high-quality subset of data for more efficient or effective model training. The key distinction lies in the availability of labels or pairings: active learning selects data points *before* annotation, whereas data selection assumes a fully labeled or paired dataset. From this perspective, active learning is strictly more challenging, as it must operate without access to labeling or pairing information.

Data selection methods have been shown to reduce training cost (Schreiber et al., 2020; Mindermann et al., 2022; Sorscher et al., 2022; Yang et al., 2022; Shen et al., 2024), and in some cases even improve performance by removing duplicated or noisy data (Lee et al., 2021; Tirumala et al., 2023;

Xia et al., 2024). In the multimodal domain, researchers have proposed metrics such as CLIPScore (Hessel et al., 2021) and its extensions (Wang et al., 2024a;b; Joshi et al., 2024) to evaluate the quality of image-ext pairs, enabling filtering of low-quality examples for data selection. While CLIPScore-based filtering has proven useful for multimodal data selection (Schuhmann et al., 2021), it assumes access to pre-paired multimodal data and is therefore unsuitable for our setting with *unaligned* modalities.

## A.2 SUPPORTING RESULTS FROM SECTION 3

**Proposition 1.** *The per-round data acquisition complexity of Algorithm 1 is upper bounded by* $O(B_C \cdot |\mathcal{D}_t| \cdot |\mathcal{S}_{t-1}|)$, *resulting in an overall complexity of* $O(T^2 \cdot B \cdot B_C \cdot |\mathcal{D}|)$.

*Proof.* We analyze the data acquisition complexity of Algorithm 1 as follows.

- *Per-round complexity.* We analyze the runtime of each major step in the algorithm:

  - *Line 5.* The main computational cost arises from evaluating Eq. (1), which involves iterating over both $\mathcal{D}_t$ and $\mathcal{S}_{t-1}$. The resulting runtime is upper bounded by $O(|\mathcal{D}_t| \cdot |\mathcal{S}_{t-1}|)$.
  - *Line 6.* For coreset construction, we use the greedy approximation algorithm in Algorithm 2. Each iteration for selecting a point and updating the coreset takes $O(|\mathcal{D}_t| \cdot |\mathcal{S}_{t-1}|)$ time. Repeating this process $B_C$ times gives an overall runtime of $O(B_C \cdot |\mathcal{D}_t| \cdot |\mathcal{S}_{t-1}|)$.
  - *Lines 7–10.* These steps perform uncertainty-based selection. Computing uncertainty scores over $B_C$ coreset candidates against the other modality requires $O(B_C \cdot |\mathcal{D}_t|)$ operations.

  Summing the contributions from each step, the per-round data acquisition complexity is upper bounded by
  $$O(B_C \cdot |\mathcal{D}_t| \cdot |\mathcal{S}_{t-1}|).$$

- *Overall complexity.* Since $|\mathcal{S}_{t-1}| = B \cdot (t-1)$, the total complexity over $T$ rounds is
  $$O\left(B_C \cdot \sum_{t=1}^{T} |\mathcal{D}_t| \cdot B \cdot (t-1)\right) = O(T^2 \cdot B \cdot B_C \cdot |\mathcal{D}|),$$
  where we use $|\mathcal{D}_t| \leq |\mathcal{D}|$ for all $t$.

$\square$

**Remark 1.** *Faster implementation of Line 6 in Algorithm 1. To improve efficiency, Line 6 (using Algorithm 2) can be implemented using a distance caching strategy. Initially, we compute the minimum distances between all candidate points in* $\mathcal{D}_t$ *and the selection set* $\mathcal{S}_{t-1}$, *incurring a runtime of* $O(|\mathcal{D}_t| \cdot |\mathcal{S}_{t-1}|)$. *For subsequent iterations in the greedy selection process, we only need* $O(|\mathcal{D}_t|)$ *operations to update the cache and select the next point. Thus, the overall runtime of Line 6 is improved to* $O\left((B_C + |\mathcal{S}_{t-1}|) \cdot |\mathcal{D}_t|\right)$.

## A.3 OTHER DETAILS FOR EXPERIMENTS

### A.3.1 ADDITIONAL DETAILS AND BASELINES

**Computing resources.** All experiments are implemented in PyTorch (Paszke et al., 2019), with parts of the codebase adapted from DataComp (Gadre et al., 2024) and OpenCLIP (Ilharco et al., 2021). Experiments are conducted on a single NVIDIA RTX 6000 Ada GPU.

**Additional details on datasets.** We use the 2017 release of MS-COCO, which contains approximately 118K training images (train split) and 5K validation images (val split).[4] Since only these two splits provide captions, we use the train split for training and the val split for testing. Each image in MS-COCO is paired with five captions; in our experiments, we use the first caption for each image.

---

[4] https://cocodataset.org

**Baselines.** We provide full implementations of the two baseline methods used in our experiments: CORESET (Algorithm 3) and UNCERTAINTY (Algorithm 4). Both are adapted from diversity-based and uncertainty-based active learning algorithms originally developed for unimodal settings.

In the multimodal setting, Algorithm 3 randomly selects a modality and then constructs a coreset within that modality using a greedy algorithm. Algorithm 4 computes margin-based uncertainty scores in both directions (text $\rightarrow$ image and image $\rightarrow$ text), and selects the top-$B$ most uncertain instances for multimodal annotation.

To extend these baselines to the streaming setting, we adopt the same strategy as in Algorithm 1: line 4 in both Algorithm 3 and Algorithm 4 is replaced with the current batch of stream data $\mathcal{D}_t$, while the remainder of each algorithm is left unchanged.

For consistency, we use Euclidean distance as the default metric in Algorithm 1, Algorithm 3, and Algorithm 4.

---

**Algorithm 3** Multimodal Coreset Selection

---

**Input:** Unaligned multimodal dataset $\mathcal{D} = \{\mathcal{D}^v, \mathcal{D}^l\}$, number of iterations $T$, per-round selection size $B$.
1: Initialize multimodal model $\phi_0 = \{\phi_0^v, \phi_0^l\}$ with random or pretrained weights.
2: Initialize the annotation set $\mathcal{S}_0 = \emptyset$.
3: **for** $t = 1, \cdots, T$ **do**
4:     Consider unaligned data pool $\mathcal{D}_t := \mathcal{D} \setminus \mathcal{S}_{t-1}$.
5:     Randomly select a modality $k_t \in \{v, l\}$.
6:     **for** $c = 1, \cdots, B$ **do**
7:         $z_u = \arg\max_{z_i \in \phi(\mathcal{D}_t^{k_t})} \min_{z_j \in \phi(\mathcal{S}_{t-1}^{k_t})} \mathsf{dist}(z_i, z_j)$.
8:         Annotate $z_u$ and add $(z_i^v, z_i^l)$ to $\mathcal{S}_t$.
9:     Train multimodal model $\phi_t = (\phi_t^v, \phi_t^l)$ on the updated annotation set $\mathcal{S}_t$.
**Output:** Actively trained multimodal model $\phi_T = (\phi_T^v, \phi_T^l)$.

---

**Algorithm 4** Multimodal Uncertainty-based Data Selection

---

**Input:** Unaligned multimodal dataset $\mathcal{D} = \{\mathcal{D}^v, \mathcal{D}^l\}$, number of iterations $T$, per-round selection size $B$.
1: Initialize multimodal model $\phi_0 = \{\phi_0^v, \phi_0^l\}$ with random or pretrained weights.
2: Initialize the annotation set $\mathcal{S}_0 = \emptyset$.
3: **for** $t = 1, \cdots, T$ **do**
4:     Consider unaligned data pool $\mathcal{D}_t := \mathcal{D} \setminus \mathcal{S}_{t-1}$
5:     For each modality $k \in \{v, l\}$ and for each data $x_i^k \in \mathcal{C}_t^k$, compute its margin score $u(x_i^k) := w_{(1)}^i - w_{(2)}^i$, which serves as an uncertainty measure. Here $w^i \in \mathbb{R}^{|\mathcal{D}_t^m|}$ denotes the vector of similarity scores between $x_i^{k_t}$ and all unaligned features in the other modality $m := \{v, l\} \setminus \{k\}$, and $w_{(j)}^i$ denotes the $j$-th largest entry of $w^i$. // Calculate the margin scores from both directions.
6:     Select $B$ data points with smallest margin scores with respect to $\{x_i^v\}_{i=1}^{|\mathcal{D}_t^v|} \cup \{x_j^l\}_{j=1}^{|\mathcal{D}_t^l|}$, annotate them, and update $\mathcal{S}_t \leftarrow \mathcal{S}_{t-1} \cup \{(x_i^v, x_i^l)\}_{i=1}^B$. // If, within the top-$B$ selection, a selected image feature is paired with another selected language feature, then continue the selection process until getting $B$ annotated pairs.
7:     Train multimodal model $\phi_t = (\phi_t^v, \phi_t^l)$ on the updated annotation set $\mathcal{S}_t$.
**Output:** Actively trained multimodal model $\phi_T = (\phi_T^v, \phi_T^l)$.

---

### A.3.2 HYPERPARAMETER SETTINGS

Table 4: Hyperparameter settings for the ColorSwap dataset using CLIP model.

| Hyperparameters | CLIP-B32 | CLIP-L14 |
|---|---|---|
| Epochs | 50 | 50 |
| Batch Size | 70 | 35 |
| Optimizer | AdamW | AdamW |
| Weight Decay | 0.1 | 0.1 |
| Learning Rate | $2 \times 10^{-5}$ | $1 \times 10^{-5}$ |
| $B_C$ | $2.5B$ | $2.5B$ |

Table 5: Hyperparameter settings for the ColorSwap dataset using SigLIP and LiT.

| Hyperparameters | SigLIP-B16 | SigLIP-L16 | LiT-L14 |
|---|---|---|---|
| Epochs | 80 | 80 | 50 |
| Batch Size | 70 | 35 | 35 |
| Optimizer | AdamW | AdamW | AdamW |
| Weight Decay | 0.1 | 0.1 | 0.1 |
| Learning Rate | $2 \times 10^{-5}$ | $1 \times 10^{-5}$ | $1 \times 10^{-5}$ |
| $B_C$ | $1.5B$ | $2.0B$ | $2.5B$ |

Table 6: Hyperparameter settings for the MS-COCO and DataComp datasets using CLIP model.

| Hyperparameters | MS-COCO | DataComp |
|---|---|---|
| Epochs | 10 | 1 |
| Batch Size | 256 | 512 |
| Optimizer | AdamW | AdamW |
| Weight Decay | $1 \times 10^{-4}$ | 0.2 |
| Learning Rate | $1 \times 10^{-5}$ | $6.25 \times 10^{-5}$ |
| $B_C$ | $2B$ | $2.5B$ |

Tables 4 to 6 list the hyperparameters used in our experiments across datasets and model variants. For ColorSwap and MS-COCO, we initialize models from pretrained weights and study multimodal active learning in the *finetuning* regime. For DataComp, we initialize models from random weights to evaluate multimodal active learning in the *pretraining* regime.

### A.3.3 ADDITIONAL EXPERIMENTAL RESULTS

We present additional experimental results in Figs. 5 to 7. The conclusions from the main results in Section 4.2 remain consistent across different CLIP variants and model sizes: Algorithm 1 continues to outperform all baselines.

### A.3.4 SUPPORTING FIGURES FOR ADDITIONAL ANALYSES

In Fig. 8, we generate a t-SNE visualization on the ColorSwap dataset using image-modality data selected by Algorithm 1 and by UNCERTAINTY.

### A.4 THE USE OF LARGE LANGUAGE MODELS (LLMS)

LLMs were used to polish the writing of this paper.

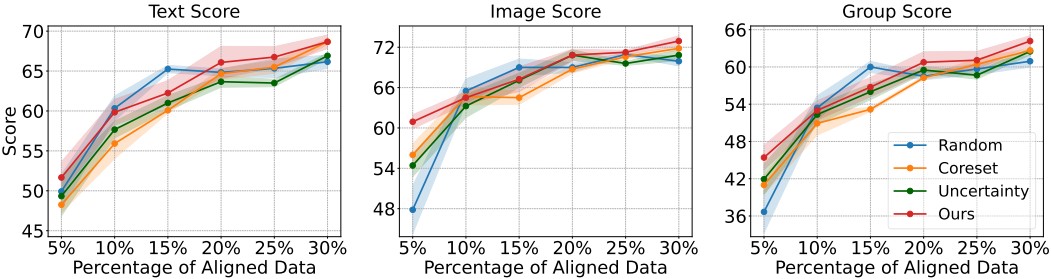

Figure 5: Results of pool-based multimodal active learning on the ColorSwap dataset with CLIP-L14. We report text score (*left*), image score (*middle*), and group score (*right*) as learning progresses.

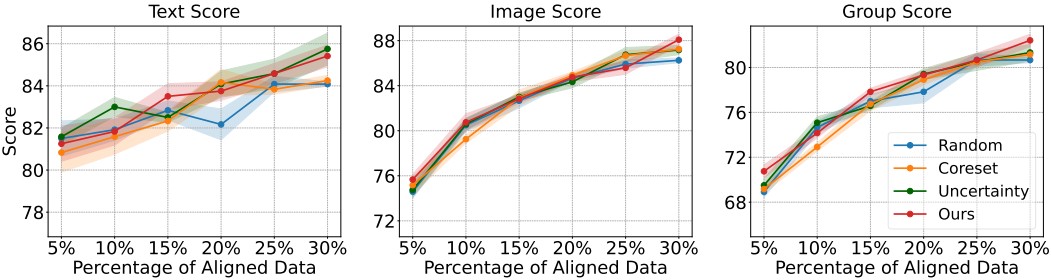

Figure 6: Results of pool-based multimodal active learning on the ColorSwap dataset with SigLIP-L16. We report text score (*left*), image score (*middle*), and group score (*right*) as learning progresses.

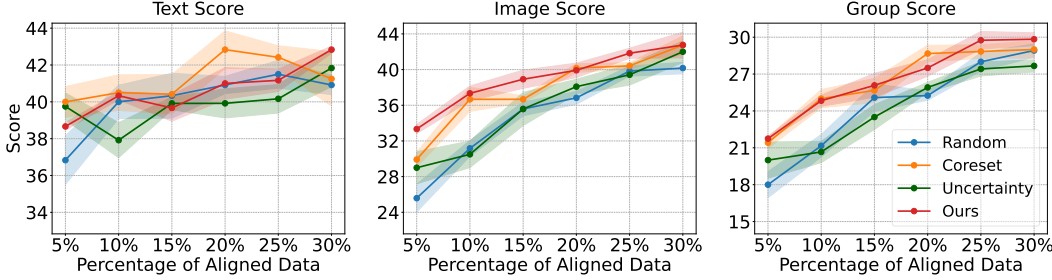

Figure 7: Results of pool-based multimodal active learning on the ColorSwap dataset with LiT-L14. We report text score (*left*), image score (*middle*), and group score (*right*) as learning progresses.

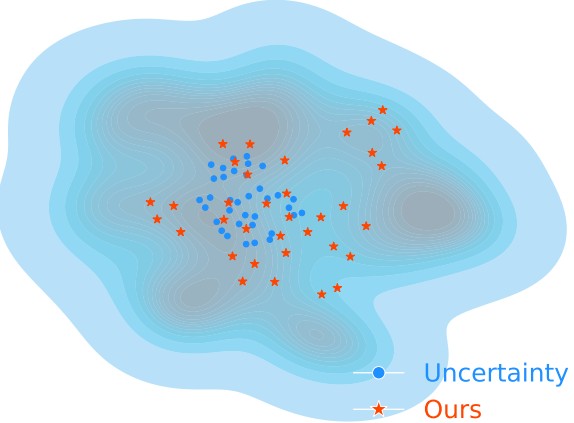

Figure 8: t-SNE visualization of image-modality embeddings from the ColorSwap dataset, comparing our method (Algorithm 1) with an uncertainty-based baseline (UNCERTAINTY). Points selected by the baseline are shown as blue circles, while points selected by our method are shown as red stars. Blue density contours represent the distribution of all data. Compared to the uncertainty-based method, our approach selects samples that not only capture uncertain regions but also exhibit greater diversity across the embedding space.

