# OpenReview forum: "Towards Multimodal Active Learning: Efficient Learning with Limited Paired Data"
_ICLR.cc/2026/Conference — ICLR 2026 Conference Withdrawn Submission_

### Official Review · Reviewer_WvET · 2025-10-18

**Soundness:** 2
**Presentation:** 3
**Contribution:** 2
**Rating:** 4
**Confidence:** 4

**Summary:**

This papre introduces a novel task, i.e., multimodal active learning with unaligned data, which is extremely challenging for current methods of multimodal training based on paired data. To address such problem, this paper concentrates on the two distinctive challenges here (Bidirectional alignment and Large cross-modal candidate space), and proposes the three-step annotation approach, integrating both diversity and uncertainty to the selection process as well as achieving linear-time complexity.

**Strengths:**

1. The proposed task is challenging, novel and of practical significance, which I think is the most contribution of this paper. Solving this problem is an effective way to enhance the performance of multimodal models in dealing with non-aligned data in practical application scenarios while maintaining a relatively low cost.
2. The paper writing and structure organization is good.
3. The two summarized challenges are reasonable and precise.

**Weaknesses:**

1. This proposed three-step method is a bit too naive. First, select a modality A with poor convergence performance, conduct diversity-maximizing sample selection within this modality A, then perform the final sample selection and annotation based on the uncertainty calculated from inter-modal relationships. However, there are several issues:
1) Since the capacity of each modality varies inherently, can the convergence index calculated via sample distance truly indicate that the selected modality is the one in greater need of training?
2) After ensuring the diversity of modality A, how to guarantee the diversity of the correspondingly selected samples from other modalities?
3) Are the finally obtained annotated sample pairs necessarily beneficial for training? The contribution of the multimodal data to the model is not considered during Steps 1-2. There is no guarantee that for the samples of modality A selected in Steps 1-2, there must exist another sample in other modalities that forms a sample pair with positive effects for model training. The authors should provide explanations for this part or avoid such issues in the method design.
2. Although the authors mention that the method proposed in this paper can be extended to modalities other than vision-language, this has not been demonstrated in the experiments. More modalities should be incorporated, such as audio-language, audio-video, or the three-modal dataset, vision-audio-language.
3. This paper focuses on multimodal training with unaligned data. Although there is currently no corresponding active learning method for comparison, methods training with unaligned multimodal data should be selected for comparison or integration.

**Questions:**

1. For the DataComp dataset, there is an order-of-magnitude difference between the experimental results published by the authors and those in the original DataComp paper. The authors are requested to provide an explanation for this discrepancy.
2. what is the zero-shot performance when using pre-trained CLIP or other models?

---

### Official Review · Reviewer_Y68X · 2025-10-24

**Soundness:** 3
**Presentation:** 3
**Contribution:** 2
**Rating:** 4
**Confidence:** 3

**Summary:**

- This paper addresses the problem of active learning in multimodal training.
- The paper considers the setting where the learner must not only query a label, but also query the cross-modal alignment.
- This is very hard, and classical AL methods would be quadratic complexity in the number of samples with two modalities.
- Their method proposes a linear variant that has strong experimental results.

**Strengths:**

- The results seem strong. I appreciate that they included plots and not just tables, with multiple trials sweeping across different alignment percentages to show clear trends. They ran many experiments and the method seems to outperform baselines most of the time, though the margin is small at times.
- The presentation is good. The ablation studies were insightful.
- Extending the experiments across models was robust and persuasive.

**Weaknesses:**

- This is maybe more of a question, but I don't know if I understand the motivation of the method. When would it be useful to query modality alignment and labels, particularly in multimodal training? The authors said that there are cases where this can be scarce, but in the Audio + Vision + Text setting, we have an abundance of aligned data on youtube, for example. They said that it might be relevant for the medical domain, but they didn't run experiments on that domain that I can tell.
- Relation to broader active learning works: what about this is inherently multimodal? For what isn't inherently multimodal, are there toy experimental settings that relate to the broader theory of active learning that would be illustrative of the method? Having a firmer grounding in the theory of the method would be helpful.

**Questions:**

See weaknesses.

---

### Official Review · Reviewer_Ygq9 · 2025-10-30

**Soundness:** 2
**Presentation:** 2
**Contribution:** 3
**Rating:** 2
**Confidence:** 3

**Summary:**

The paper introduces multimodal active learning with unaligned data, where the learner must actively acquire cross-modal alignments instead of labels on pre-aligned pairs. This setting aims to address the problem where unimodal features are abundant but high-quality alignment is costly. The authors introduce a 3-step algorithm that performs modality selection, coreset construction, and uncertainty-based selection to provide a set of samples to be aligned and labeled at an annotation cost.

**Strengths:**

- Novel and Practical Problem Formulation: First to study active learning in the challenging setting of unaligned multimodal data.

- Algorithmic Contribution: Three-step algorithm incorporating modality selection to balance coverage, diversity-based coreset construction for efficiency, and uncertainty-driven sample selection for informativeness.​

- Diverse Applicability: Seamless applicability to both pool-based and streaming data settings with careful computational complexity analysis.​

- Margin Score Introduction: The use of margin scores (the difference in similarity of one modality’s element to the most similar item in the next modality and the second most similar item) for understanding uncertainty in alignment is interesting and innovative, with many potential applications. Is this something based on previous works?

- Detailed Analyses and Ablations: Shows the effectiveness of the modality selection strategy, as well as the margin score validity analyzed on pseudo-alignments, demonstrating strong uncertainty signal.​ Parameter sensitivity experiments show stable performance across the coreset size hyperparameter.

**Weaknesses:**

- Unclear Problem Motivation and Practical Context: The paper fails to establish clear practical scenarios where multimodal active learning with unaligned data would be applied. Would a practical scenario require a human-in-the-loop annotation process? Otherwise how would the system query for and receive alignment labels in practice​ The relationship to existing systems like CLIP is slightly misleading, as CLIP uses already-paired (albeit potentially noisy) web data, not unpaired data requiring active alignment discovery
- Confusing Bidirectional Alignment Concept: The notion of bidirectional alignment (lines 49-51) lacks a clear explanation and practical examples throughout the paper. The distinction between vision-to-language versus language-to-vision alignment is not sufficiently clarified with concrete illustrations.
- Ambiguous Data Setting: The assumption that datasets Dv and Dl contain corresponding 1:1 pairs that are simply unknown is poorly motivated. It remains unclear when this scenario occurs in practice beyond treating one modality as input and another as label, which reduces to traditional active learning
- Vague Experimental Validation: Experimental details are insufficiently described, particularly for tasks like image-to-text retrieval on MS-COCO. The inputs, outputs, and evaluation protocols for ColorSwap, MS-COCO, and DataComp experiments lack clarity​. Baseline methods are inadequately described in the main paper, requiring readers to infer their functionality. I would suggest prioritizing including their description in the main paper and finding other parts to move to the appendix (such as Algoirthm 2).
- Underexplored Modality Representation Concept: The notion of "underrepresented modality" (lines 203-204) is poorly explained without concrete examples. The practical meaning of distance-based modality representation in the context of alignment selection remains unclear​
- Limited Analysis of Annotation Costs: Despite introducing annotation costs (lines 83-85), the framework assumes unit costs throughout without addressing when and why annotation difficulty varies by modality or alignment direction​.
- Limited Exploration of Noisy or Imperfect Alignment Queries: The approach assumes queried alignments are always correct; robustness to noisy or ambiguous annotations is not explicitly addressed.​
- Focused primarily on Vision-Language Pair Data: While generalization to more modalities is claimed, empirical validation on multimodalities beyond vision-language is not demonstrated.​

**Questions:**

Clarifications:
- Bidirectional Alignment Examples: Can you provide a concrete example demonstrating how "different alignment directions lead to entirely different annotation sets" as claimed in lines 131-132? When would the same visual instance align to different textual instances depending on direction?​ For example, how is aligning an image of a dog and the text “dog” different from aligning the text “dog” to an image of a dog?
- Cross-Modal Candidate Space: Why must the learner evaluate all instances in the opposite modality (lines 133-138)?
- Modality Underrepresentation: How can one modality be underrepresented when selecting any instance for annotation, also provides its corresponding pair in the other modality? What practical scenarios does distance-based underrepresentation capture?​ From equation 1, it seems underrepresentation is when the distances in the latent space of one modality to other instances of that modality are farther, but what does that mean/represent? Is this talking about which classes or concepts are underrepresented?
- Application Scenarios: What are specific use cases for pool-based versus stream-based multimodal active learning? Why is the streaming discussion relevant?​
- Computational Complexity Claims: What are the actual values of T, B, Bc, and |D| in your experiments? The claim that T²BBc < |D| seems questionable. For 100 samples using 25 for active learning training with 5 batches of 5, wouldn't T²BBc = 625 >> |D| = 100? So it seems like this approach would be less efficient than D^2. ​

Though Experiments/Extensions
- Variable Annotation Costs: Can the framework incorporate realistic scenarios where annotation effort varies significantly across modalities or query types?​
- Noise Robustness: How does the method handle uncertain or partially correct alignment annotations from human annotators?​
- Beyond Contrastive Models: What modifications are needed to extend this framework beyond CLIP-style contrastive learning architectures?​
- Generative Model Extension: How might this approach be adapted for multimodal generative models, and what technical challenges would arise in that context?

---

### Official Review · Reviewer_xt8T · 2025-11-03

**Soundness:** 2
**Presentation:** 2
**Contribution:** 2
**Rating:** 4
**Confidence:** 3

**Summary:**

This paper introduces a novel and practical problem setting: multimodal active learning with unaligned data. The paper identifies two key challenges in this setting: bidirectional alignment and a large cross-modal candidate space. To address this, the authors propose an efficient algorithm that integrates modality-aware diversity and cross-modal uncertainty in a three-step process, achieving linear-time complexity. Experiments on benchmarks like ColorSwap, MS-COCO, and DataComp have been conducted to demonstrate the performance.

**Strengths:**

1. The paper introduces a novel problem formulation of unaligned multimodal active learning, important in cases where alignment serves as the primary cost bottleneck.

2. The paper presents empirical validation across multiple datasets under both pool-based and streaming-based settings. Furthermore, the ablation studies provide evidence supporting the effectiveness of the margin score.

**Weaknesses:**

1. The paper lacks theoretical justification or quantitative analysis for each component of the proposed framework, including modality selection, coreset construction, and the uncertainty measure.

2. The experimental results (e.g., Figures 1 and 3) are not consistently better than baselines, and the paper does not clearly interpret the metrics used. Should the proposed method perform consistently better across all percentages of aligned data?

**Questions:**

1. The paper employs a max-min distance metric to assess coverage for modality selection and coreset construction. Could the authors discuss the rationale behind this choice compared to other potential methods, such as clustering-based (e.g., k-means cluster centroids) or density-based approaches? What are the theoretical or empirical advantages of the max-min strategy in this specific context of unaligned multimodal data?

2. The margin score is used as the uncertainty measure. Why is this specific measure particularly effective for quantifying uncertainty in a cross-modal matching task? Were other uncertainty metrics, such as least confidence or the entropy of the similarity distribution, considered? What are the relative advantages of the margin score in this setting?

3. In Figures 1 and 3, the proposed method is occasionally outperformed by a baseline at certain budget levels. How should these performance drops be interpreted?

---

### Note · Authors · 2025-11-14

**Comment:**

After careful consideration, we have decided to withdraw our submission from the conference. We appreciate the reviewers’ time and feedback, which will help us further refine and strengthen our work.

**Withdrawal Confirmation:**

I have read and agree with the venue's withdrawal policy on behalf of myself and my co-authors.